# Oxidative Stress-Associated Alteration of circRNA and Their ceRNA Network in Differentiating Neuroblasts

**DOI:** 10.3390/ijms252212459

**Published:** 2024-11-20

**Authors:** Ebrahim Mahmoudi, Behnaz Khavari, Murray J. Cairns

**Affiliations:** 1School of Biomedical Sciences and Pharmacy, The University of Newcastle, Newcastle, NSW 2308, Australia; 2Precision Medicine Research Program, Hunter Medical Research Institute, Newcastle, NSW 2305, Australia

**Keywords:** circRNAs, ceRNA network, interaction axes, oxidative stress, neuronal differentiation, neurodevelopmental disorders, SH-SY5Y

## Abstract

Oxidative stress from environmental exposures is thought to play a role in neurodevelopmental disorders; therefore, understanding the underlying molecular regulatory network is essential for mitigating its impacts. In this study, we analysed the competitive endogenous RNA (ceRNA) network mediated by circRNAs, a novel class of regulatory molecules, in an SH-SY5Y cell model of oxidative stress, both prior to and during neural differentiation, using RNA sequencing and in silico analysis. We identified 146 differentially expressed circRNAs, including 93 upregulated and 53 downregulated circRNAs, many of which were significantly co-expressed with mRNAs that potentially interact with miRNAs. We constructed a circRNA–miRNA–mRNA network and identified 15 circRNAs serving as hubs within the regulatory axes, with target genes enriched in stress- and neuron-related pathways, such as *signaling by VEGF*, *axon guidance*, *signaling by FGFR*, and the *RAF/MAP kinase cascade*. These findings provide insights into the role of the circRNA-mediated ceRNA network in oxidative stress during neuronal differentiation, which may help explain the regulatory mechanisms underlying neurodevelopmental disorders associated with oxidative stress.

## 1. Introduction

Oxidative stress is a common phenomenon induced by an imbalance between the production of reactive oxygen species (ROS) and their clearance by the antioxidant system. This process causes significant molecular and cellular damage to tissues, particularly in the brain, which is highly susceptible to oxidative stress due its elevated level of oxygen consumption and lipid-rich content [1]. A highly oxidative environment affects neuronal structural integrity and disrupts their optimal function, potentially leading to pathological conditions in the human brain [2].

Emerging evidence suggests that oxidative stress may be an important contributor to a host of neurodegenerative and neuropsychiatric disorders, such as Alzheimer’s Disease (AD), Parkinson’s Disease (PD), schizophrenia, bipolar disorder, and depression, where it impairs critical processes including cognition and neurotransmission [1,3,4]. In psychiatric conditions, these mechanisms may be mediated via the inflammatory response of the brain, which frequently appears in such disorders and may play an important role in their pathophysiology [5]. Perturbation of antioxidant enzymes and elevated levels of free radicals have been observed in bipolar disorder, schizophrenia, and major depression [6,7,8]. Studies have shown that using antioxidants, such as vitamins C and E, EGb, and ginkgo, improved clinical symptoms in schizophrenia [9], supporting an association between oxidative stress and the disorder’s pathophysiology.

Given the potential involvement of oxidative stress in the development of various diseases and its detrimental effects on the genome [10], there has been increasing effort to explore the molecular consequences of oxidative conditions. This includes gene expression studies, which have revealed significant transcriptomic responses to oxidative stress [11,12,13]. In our recent work, we observed alterations in many genes in response to oxidative stress, particularly in pathways associated with psychiatric disorders and immunity-related clusters [13]. Additionally, associated changes in non-coding RNA (ncRNA), such as microRNA (miRNA) and long non-coding RNA (lncRNA), may mediate the effects of oxidative stress [14,15], and thus play a critical role in its pathophysiological consequences. For example, our lab has demonstrated that oxidative conditions induce dysregulation of numerous miRNAs involved in psychiatric disorders, including 12 miRNAs originating from DLK1-DIO3, a schizophrenia-associated locus [16].

Circular RNAs (circRNAs) are a class of ncRNAs formed through the back-splicing of pre-mRNA transcripts [17]. CircRNAs are evolutionarily conserved and stably expressed in almost all mammalian cells and tissues, and they appear, at least in some cases, to modulate the activity of target nucleic acids, including miRNA and mRNA [18]. These transcripts, primarily present in the cytoplasm, are usually derived from coding genes and in some cases contain open reading frames that support translation [19,20]. CircRNAs are implicated in various cellular processes, including proliferation, differentiation, inflammatory and immune responses, apoptosis, and development [21,22]. Increasing evidence indicates that circRNAs are associated with a range of human diseases, including neuropsychiatric and neurodegenerative disorders such as schizophrenia, AD, autoimmune diseases, cancer, and cardiovascular diseases [23,24,25,26]. Mechanistic studies have demonstrated the regulatory function of circRNAs by fine-tuning mRNA at the transcriptional and post-transcriptional levels [17]. These molecules have been shown to act as competitive endogenous RNAs (ceRNAs) that bind miRNAs and attenuate their target gene-silencing activity. For example, a circRNA-associated ceRNA network is known for CDR1as, a brain-enriched circRNA that sponges miR-7, resulting in upregulation of the miRNA gene targets of Fos, Nr4a3, Irs2, and Klf4 [27,28,29].

Given the association of oxidative stress as a significant risk factor for neurological disorders, a deep understanding of the molecular basis, in particular the regulatory systems that mediate oxidative stress-induced changes, seems essential to uncovering their mechanisms in the pathogenesis of these disorders. This understanding may also aid in developing novel diagnostic agents or therapeutic targets for treating pathological conditions characterised by oxidative features, such as neurodevelopmental diseases. CircRNAs serve as regulatory molecules that have been implicated in oxidative stress-associated pathological processes and disorders. We therefore hypothesised that circRNAs and their ceRNA network may mediate molecular perturbations arising from oxidative stress. To investigate this, we reanalysed our total RNA expression dataset using a circRNA-oriented pipeline and detected thousands of circRNAs, many of which were found to be associated with oxidative stress. Our bioinformatic analysis suggested these circRNAs have the capacity to modulate the expression of target genes through circRNA–miRNA–mRNA interactions.

## 2. Results

### 2.1. Oxidative Stress Models and Profiling the circRNAome

In order to explore the circRNA-mediated impacts of oxidative stress and their critical role in neuronal development, we established two models using human SH-SY5Y neuroblast cultures: (i) pre-differentiation oxidative stress (PD-OS), in which oxidative stress was induced prior to neuronal differentiation, and (ii) during-differentiation oxidative stress (DD-OS), where oxidative stress was induced during differentiation. We determined transcriptome-wide profiles of circRNA in the two oxidative stress models by conducting RNA sequencing using total RNA depleted of ribosomal RNA. To predict and quantify the circRNAs, we used CIRIquant, a recent pipeline with high precision and reliability [30]. A total of over 51,000 circRNAs were initially identified in all the samples, but to ensure high confidence in the circRNAs, we applied a cutoff minimum of two junction reads in at least half of the samples in each oxidative stress model. This resulted in detection of 1910 and 1282 circRNAs in PD-OS and DD-OS, respectively (Appendix A). We used these circRNAs for all the analyses to ensure the reliability of the results. A flowchart of the experimental design and circRNA detection process is illustrated in Figure 1.

### 2.2. CircRNAs Are Differentially Expressed in Response to Oxidative Stress

To explore the circRNAome response to oxidative stress in neuronal differentiation, we performed differential expression analysis in PD-OS and DD-OS. The results showed that a total of 102 and 45 circRNAs were altered in PD-OS and DD-OS, respectively (*p* < 0.05, FC > 2; Appendix A). This included circTCONS_l2_00000968, circSCARNA10, circCCDC90B, and circC7orf44, which survived multiple testing corrections in PD-OS, and circSIPA1L3 and circCANX in DD-OS. As illustrated in Figure 2a,b, about two thirds of the circRNAs were upregulated following the induction of oxidative stress: 65 circRNAs in PD-OS and 28 circRNAs in DD-OS. To further validate these observations, we examined the expression of the top 11 differentially expressed circRNAs (7 from PD-OS and 4 from DD-OS) using qRT-PCR (Figure 2c), and confirmed that the circRNA expression inferred from sequencing data correlated with qRT-PCR (Spearman’s rank, ρ = 0.74, *p* = 0.013) (Figure 2d). We also noticed that the circRNA expression response to PD stress was distinct from that of DD stress, as there was only one circRNA, chr4:73956384|73958017 (circANKRD17), common between the two stress conditions. In order to investigate the potential mechanisms of action of the altered circRNAs, we performed Gene Ontology (GO) analysis for their parental genes. We found a significant enrichment of 27 biological processes in PD-OS, including *embryo development*, *brain development*, *neural tube development*, *forebrain development*, *head development*, and *embryonic morphogenesis* (Figure 2e; Appendix A). In DD-OS, however, there was no significant enrichment observed.

### 2.3. Identification of circRNA-Mediated ceRNA Network

Recent studies suggest that circRNAs containing MREs can competitively sponge miRNAs and inhibit them from binding to target sites within mRNA transcripts, thereby fine-tuning the expression of mRNA [27,28,29]. We therefore sought to characterise the circRNA-associated ceRNA regulatory network in oxidative stress. To this end, we integrated the expression profiles of circRNA with mRNA expression obtained from reanalysing our previous dataset [13], as well as miRNA expression results recently published by our lab [16]. As the potential regulatory relationships are reflected in a positive correlation of circRNAs with their mRNA targets, we determined the DE mRNA in oxidative stress (211 in PD-OS and 224 in DD-OS) and then examined their co-expression with the differentially expressed (DE) circRNAs across the samples in each model. This analysis revealed 2301 circRNA–mRNA pairs in PD-OS and 367 pairs in DD-OS that showed a positive expression correlation (r > 0.90, adjusted *p* < 0.05; Appendix A). To construct a ceRNA network of circRNA–miRNA–mRNA, we predicted the miRNA binding sites for each correlated circRNA–mRNA pair. Pairs where both elements shared a common miRNA that was also expressed in the samples were included (Figure 3a). The resulting network identified 454 axes and 245 nodes in PD-OS, which included 301 circRNA–miRNA and 265 miRNA–mRNA interactions, and 122 axes and 103 nodes in DD-OS, including 95 circRNA–miRNA and 84 miRNA–mRNA interactions (Appendix A). The ceRNA networks of circRNA–miRNA–mRNA are illustrated in Figure 3b for PD-OS and 3c for DD-OS.

Intriguingly, we noticed that more than half of the miRNAs predicted to be sponged by differentially expressed circRNAs were previously reported to be dysregulated in the brain and peripheral tissues of psychiatric patients [31]. This included 66/114 miRNAs in PD-OS and 36/60 miRNAs in DD-OS, with 31 miRNAs affected by both conditions (Figure 4a and Appendix A). A Fisher’s exact test revealed a statistically significant enrichment of sponged miRNAs in psychiatric disorders for both conditions (*p*-value < 0.001) (Figure 4b,c).

### 2.4. Characterisation of Hub circRNAs in the ceRNA Networks

To determine the central regulatory elements involved in the oxidative stress process, we ranked the circRNA nodes in the network using CytoHubba and found 10 and 5 key circRNAs with highest connectivity in the ceRNA networks of PD-OS and DD-OS, respectively (Appendix A). A regulatory subnetwork was constructed for each oxidative stress model, as shown in Figure 5. The PD-OS subnetwork was composed of 255 circRNA–miRNA–mRNA interactions where the hub circRNAs potentially regulate 57 mRNAs by sponging 86 miRNAs (Figure 5a; Appendix A). Similarly, in DD-OS, we detected 97 circRNA–miRNA–mRNA interactions, where the hub circRNAs potentially regulate 20 DE mRNA through binding to 46 miRNAs (Figure 5b; Appendix A). Three circRNAs, circZNF292, circBPTF, and circFAM193B, were the top regulatory hubs in PD-OS, while circZC3H13, circSFMBT2, and circCPT1A were the top hubs in DD-OS, suggesting potential significant roles for these circRNAs in the biology of stress. Examples of the observed regulatory axes are circZNF292/miR-222-3p/C11orf88, circBPTF/miR-138-5p/AJAP1, and circZC3H13/miR-15a-5p/PPM1H (Figure 5).

### 2.5. Functional Enrichment Analysis of the ceRNA Subnetworks

To glean insight into the biological functions of the key axes of the ceRNA network, we performed GO and pathway enrichment analysis for the target genes. As shown in Figure 6a, the target genes in the PD-OS network were significantly enriched with biological themes related to cell adhesion, development, neuronal processes, and response to stress (FDR< 0.05). We also found significant enrichment of the genes in many pathways, including *signaling by VEGF*, *axon guidance*, *signaling by FGFR*, *Interleukin-2 signaling*, *PI3K/AKT activation*, *negative regulation of PI3K/AKT network*, and *RAF/MAP kinase cascade*, among others, in the PD-OS regulatory network (FDR < 0.05) (Figure 6b). The full list of the enriched GO categories and pathways is provided in Appendix A. The enrichment analysis for the DD-OS network showed that the *ID signaling pathway* was enriched, while no biological processes reached the statistical significance after multiple testing correction.

## 3. Discussion

Oxidative stress is thought to play a role in the onset of neurodevelopmental disorders and may represent a major environmental factor driving the genomic instability and gene expression deregulation observed in these disorders. In this study, we explored the impact of oxidative stress on circRNA expression and its ceRNA network in differentiating SH-SY5Y neuroblast cells. We observed 102 and 45 differentially expressed circRNAs in cells experiencing stress conditions prior to differentiation (PD-OS) and during differentiation (DD-OS), respectively. Interestingly, eight of these circRNAs were previously found by our group to be differentially expressed in the postmortem cortical grey matter of schizophrenia subjects [23]. These include circXRN2, circKIAA0368, circSTK39, circTTBK2, circDLG1, and circARFGEF1 in PD-OS; and circATP8A1 and circBARD1 in DD-OS.

We observed distinct profiles between the PD-OS and DD-OS conditions, with only one circRNA, circANKRD17, in common, suggesting that the circRNAomes response to oxidative stress in early neural progenitor cells may be different from that in cells undergoing neuronal differentiation. Moreover, alteration of expression was substantially higher in PD-OS, indicating that exposure to stress may be more critical during the early stages of neural development. Our analysis confirmed that the parental genes of the altered circRNAs were significantly enriched in developmental clusters such as *embryo development, brain development, head development,* and *forebrain development* in PD-OS.

Evidence emerging from other studies has demonstrated that circRNA with miRNA recognition elements can function as competing RNA molecules by sponging their target miRNA and inhibiting them from binding to the mRNA targets [27,28,29]. We performed bioinformatics analysis to explore putative circRNA-related ceRNA regulatory networks, including circRNA, miRNA, and mRNA, in response to oxidative stress. While we integrated our previous miRNA expression results [16] directly into the interaction analyses, we reanalysed the total RNA sequencing using an effective approach, CIRIquant [30], to detect and quantify both linear and circular transcripts. We found that many DE circRNAs were significantly co-expressed with DE mRNA (positive correlation), suggesting a regulatory relationship between these transcripts. Many of these circRNA–mRNA correlation pairs were predicted to interact with common miRNA molecules, which supported the existence of ceRNA networks where circRNA and miRNA compete for the target mRNA. Using in silico interaction analysis, we constructed a ceRNA network including 454 axes in PD-OS and 122 axes in DD-OS (Figure 3).

To further characterise the key regulatory axes implicated in the oxidative stress process, we determined the hub circRNAs in the ceRNA networks and found 15 hub nodes with the highest connectivity in the networks (Figure 4). The top circRNAs include circZNF292, circBPTF, and circFAM193B in PD-OS, and circZC3H13, circSFMBT2, circCPT1A in DD-OS, suggesting these circRNAs may have critical regulatory functions in the biology of stress conditions through miRNA sponging, as shown by previous studies. For example, an isoform of circZNF292, hsa_circ_0004058, was recently reported to be significantly reduced following induction of oxidative stress in human lens epithelial HLE-B3 cells. Functional studies showed that circZNF292 overexpression significantly increased cell viability and cell cycle progression but suppressed cell apoptosis in H_2_O_2_-treated HLE-B3 cells. While H_2_O_2_ reduced the activity of antioxidant enzymes catalase and superoxide dismutase, and increased the oxidative stress marker malondialdehyde activity, circZNF292 expression significantly reduced these impacts by directly targeting miR-222-3p and regulating E2F3 expression, suggesting a role for circZNF292 in alleviating oxidative stress-induced injury [32]. This circRNA was also responsive to hypoxia in three different cancer cell lines from cervical (HeLa), breast (MCF-7), and lung (A549) cancer [33]. Similarly, it was reported to be hypoxia-induced in cultured human umbilical venous endothelial cells [34]. Another top circRNA, circBPTF (chr17:65941525|65972074), was found to be increased in bladder cancer, with its expression linked to tumour grades associated with poorer prognosis. Mechanistically, circ-BPTF was shown to inhibit miR-31-5p to allow upregulation of the oncogenic molecule RAB27A [35]. A study by Hu et al. demonstrated that circSFMBT2 (chr10:7262373|7327916) was upregulated in invasive pituitary adenomas, suggesting that the circRNA could facilitate tumour invasion by targeting an miR-15a/16-Sox-5 axis [36]. These findings are noteworthy given that oxidative stress is known to influence the progression of various cancers [37].

The expression of circFAM193B (chr5:176963359|176966148), also differentially expressed in our study, was previously altered in human induced pluripotent stem cell (hiPSC)-derived neurons, six weeks after differentiation, in patients with early-onset schizophrenia, as compared to control samples. Whilst circZC3H13 and circCPT1A have not yet been the subject of functional analysis, they may be involved in molecular mechanisms of stress through their host genes, as previously suggested [38]. This is tantalizing given that ZC3H13 protein is involved in RNA N6-methyladenosine (m6A) methylation writing. RNA m6A modification is thought to play an important role in regulating cellular response to oxidative stress [39]. The CPT1A protein is implicated in mitochondrial oxidation of fatty acids. It enhances oxidative stress and inflammation induced by ROS in liver injury [40], and also increases mitochondrial ROS while promoting antioxidant defences in prostate cancer [41].

To uncover the central regulatory elements involved in oxidative stress, we constructed ceRNA networks (Figure 5) with 255 and 97 circRNA–miRNA–mRNA axes in PD-OS and DD-OS, respectively, where the hub circRNAs associated with the expression of 57 (PD-OS) and 20 (DD-OS) DE mRNA. To further speculate on the biological function of the ceRNA regulatory networks, we conducted functional annotation and pathway analysis for the target genes. Our enrichment analysis for the PD-OS network indicated that the circRNA target genes were significantly associated with *cell adhesion*, *development*, *neurogenesis*, *axon guidance*, *neuron projection development*, as well as *response to stress* (Figure 6a). There was also a significant enrichment of the genes in various stress- and neuronal- related pathways such as *signaling by VEGF, axon guidance, signaling by FGFR, and RAF/MAP kinase cascade* (Figure 6b). These data provide insight into the molecular regulatory systems that derive critical pathways in oxidative exposure and might explain the mechanism of oxidative stress involvement in neuropathological conditions. In support of this, our previous study indicated that oxidative stress may contribute to the development of psychiatric disorders by disrupting pathways involved in synapsis, neuronal differentiation, and the immune system [13]. This is interesting given that we observed significant enrichment of circRNA-sponged miRNAs in psychiatric diseases, with 31 psychiatry-associated miRNAs common between the two exposure paradigms, including hsa-miR-17-5p, hsa-miR-181b-5p, and hsa-miR-195-5p, all of which have been found by several studies to be dysregulated in different brain regions and peripheral tissues of individuals with psychiatric disorders [31].

Our study has some limitations that warrant further investigation in future studies. Most importantly, the ceRNA networks involved in our observations need to be validated functionally through interventional loss- and/or gain-of-function studies. However, these remain challenging, as our capacity to effectively differentiate between circRNAs and cognate linear mRNAs is limited. Considering that, for many circRNAs, the same exon exists in the cognate mRNA, sequence changes at circRNAs are likely to cause unwanted changes in the expression of cognate linear RNAs, as well [42,43], although techniques such as the CRISPR-Cas13 system have been shown to offer greater efficiency and specificity compared to conventional RNAi-based approaches [43]. Furthermore, RNA pulldown assays may provide support for the interactions underpinning the ceRNA networks involved in our investigation. Proteomics and metabolomics may also provide insight into the downstream molecular changes associated with oxidative stress in the nervous system, particularly when post-transcriptional regulation of mRNA targets alter translation [44,45]. Methodologies coupling liquid chromatography with single-stage mass spectrometry (LC-MS) and nuclear magnetic resonance (NMR) spectroscopy have the potential to provide advantages over other platforms [46], through enhanced sensitivity [47] and resolution [48].

In addition, in this study, we leveraged the SH-SY5Y cell line for its low cost, ease of culture (feasibility), and reproducibility. Using alternative human neuronal models, such as embryonic stem cells, neuronal progenitor cells (NPCs), and induced pluripotent stem cells (iPSCs), offers advantages and could help confirm these results across various biological models. Therefore, repeating this study using these neural cell lines and tissue cultures will enhance our understanding of ceRNA regulation in response to oxidative stress and other environmental factors affecting neuronal development.

In summary, this study provides evidence that circRNA expression is involved in the cellular response to oxidative stress. In addition, bioinformatics analysis suggests that circRNAs emerge as a new layer of regulatory system in the stress process through associating with the ceRNA networks. Discovery of the circRNA-mediated network that responds to oxidative stress would improve our understanding of the basis of this process, and more significantly, could identify novel agents for the diagnosis or targets for therapeutic purposes in disorders with oxidative features, such as neurodevelopmental diseases. Our observation of a more significant response in the ceRNA network of PD-OS compared to the DD-OS model suggests that the most critical impacts of stress may occur before the initiation of neuron differentiation. This may indicate that neuronal progenitor cells in the fetal nervous system may be more vulnerable to stress exposure during the early stages of development. This warrants further studies, such as those conducted with in vivo animal models, to explore the prenatal impacts of oxidative stress and associated regulatory networks on the neuronal phenotypes and development of nervous system disorders.

## 4. Materials and Methods

### 4.1. Cell Culture and Differentiation

Human neuroblastoma SH-SY5Y cells were cultured at a density of 2 × 10^4^/cm^2^ in Dulbecco’s Modified Eagle’s Medium (DMEM, Burlington, VT, USA) complemented with 2 mM L-glutamine ((Hyclone, Logan, USA), 20 mM HEPES (Thermofisher, New York, NY, USA), and 10% foetal bovine serum (FBS, Bovogen Biologicals, Melbourne, Australia). The cells were maintained at 37 °C in a 5% CO_2_ with the media replaced every 2–3 days. To obtain neuronal differentiation, the immature neuroblast cells were cultured in 6-well plates, followed by treatment with 10 μM all-trans retinoic acid (ATRA) (Sigma-Aldrich, Burlington, VT, USA) the next day. The ATRA-supplemented media was refreshed after 72 h, and the differentiation protocol was terminated after 7 days of treatment with ATRA. During differentiation, the plates were covered with aluminium foil to protect them from light. The neuronal differentiation was confirmed by observing morphological and molecular changes. The cells demonstrated a neuronal phenotype, with increased growth of neurite projections and stalled cell division. In addition, we detected a significant alteration of neuronal marker genes, including TUBB3, ENO2, MAP2, MAPT, and SV2A in DD-OS (log_2_CPM values 9.6, 7.6, 8, 3.9, and 6.7, respectively) and PD-OS (log_2_CPM values 9, 7, 8.7, 4, and 6.5, respectively) conditions.

### 4.2. Inducing Oxidative Stress

As described previously [13], we designed two treatment models for oxidative stress to assess the impact of stress, both before the neuronal differentiation commenced and during the differentiation process, as follows: [1] undifferentiated cells were exposed to H_2_O_2_ (Sigma-Aldrich, Burlington, VT, USA) throughout the entire 7 days of differentiation using ATRA; we refer to this as “during-differentiation oxidative stress” (DD-OS). This condition highlights the interaction between oxidative stress and differentiation factors [2]. Undifferentiated cells were first exposed to H_2_O_2_ for 72 h and then, following removal of peroxide, treated with ATRA for 7 days to induce differentiation; we call this “pre-differentiation oxidative stress” (PD-OS). The PD-OS model allows cells sufficient time to activate adaptive mechanisms against the induced oxidative stress, indicating that the observed alterations were not merely due to a rapid increase in the concentration of ROS.

Hydrogen peroxide is commonly used to investigate the impacts of chronic oxidative stress exposure due to its high stability compared to other known reactive oxygen species, such as free radicals, superoxide anion, and hydroxyl radical [49]. To identify a suitable, non-cytotoxic concentration of peroxide that induces oxidative stress without significantly reducing cell viability, we first conducted a literature review of similar studies and then optimised the amount of H_2_O_2_ used. In a notable study, Brennand et al. treated neural progenitor cells (NPCs) with a 50 μM concentration of H_2_O_2_, and showed that this dose presented a sub-threshold environmental challenge of oxidative stress [50]. Based on this and other findings, we treated cells with increasing peroxide concentrations of 10, 20, 40, 80, and 100 μM, monitoring cellular morphology over 10 consecutive days through optical microscopy. We observed that 10 μM H_2_O_2_ had the least adverse effects; therefore, we selected it for our experiments. This choice of optimal dose was based on continuous monitoring of cell health and morphology under the microscope.

Each protocol included six samples: three H_2_O_2_ treatments and three mock treatment controls that received differentiating culture medium without H_2_O_2_. This design provided three biological replicates per treatment group for each protocol.

### 4.3. RNA Extraction and Integrity Check

Cells were lysed by 1 mL Trizol reagent (Sigma-Aldrich, Burlington, VT, USA) and lysate was collected into microcentrifuge tubes to add 200 μL of chloroform (Chem-supply, Gillman, Australia) before centrifugation at 13,000× *g* at 4 °C for 10 min according to the manufacturer’s instructions (ThermoFisher, New York, NY, USA). The resultant aqueous phase was separated from the organic phase and mixed with 80 μg glycogen (Life Technologies, Mulgrave, Australia) and 500 μL isopropanol (Chem-supply, Australia), and incubated at −20 °C overnight. Following centrifugation at 9000× *g* at 4 °C for 30 min, the supernatant was discarded, and the pellet was washed twice with 75% cold ethanol and then resuspended in nuclease-free water for storage at −80 °C. The concentration and purity of the extracted RNA was analysed by the Agilent small RNA kit and the 2100 Bioanalyzer in accordance with the manufacturer’s instructions (Agilent Technologies, Santa Clara, CA, USA). The provided system software was used for automatic calculation of the RNA integrity number (RIN) [51]. All samples had RIN values above 8.5.

### 4.4. Total RNA-Seq Library Generation and Sequencing

Library generation for total RNA-seq was performed as previously described [13]. Briefly, a total of 350 ng of total RNA was treated with DNase I (1 U/μg RNA, Thermo Scientific, New York, NY, USA), and then depleted of ribosomal RNA using the Ribo-Zero kit (Illumina, San Diego, CA, USA). The sequencing libraries were constructed using the Illumina TruSeq Stranded Total RNA Library Prep Kit according to the manufacturer’s protocol. The quality and concentrations of the constructed libraries were assessed by the High Sensitivity DNA Bioanalyzer chip (Agilent Technologies, Santa Clara, CA, USA), followed by pooling the libraries and running in paired-end mode on a NovaSeq 6000 instrument for 100 cycles.

### 4.5. Small RNA Sequencing

For a separate study [16], small RNA sequencing was performed by the Ramaciotti Centre for Genomics (UNSW, Sydney, Australia), using the QIAseq miRNA Library Kit for small RNA library preparation, and the Illumina NextSeq 500 platform (Illumina, San Diego, CA, USA) for sequencing.

### 4.6. CircRNA Detection and Expression Analysis

Sequencing reads were demultiplexed based on unique adapter sequences and then converted to FASTQ files using the “bcl2fastq” package from Illumina (https://support.illumina.com/sequencing/sequencing_software/bcl2fastq-conversion-software.html) (accessed on 20 October 2020). The reads were then quality checked using the AfterQC tool (v0.9.7) [52], such that low-quality reads were discarded and sequencing errors were corrected. The resultant clean FASTQ files were analysed by the CIRIquant package (v1.0) [30] to detect and quantify circRNA transcripts. Briefly, the reads were mapped to the reference genome (hg19) using BWA-MEM aligner [53] and unmapped reads were fed into CIRI2 (v2.0) [54] to detect back-spliced junction (BSJ) reads using the ensemble gene annotation GRCh37 (Release 19). The candidate circular reads were then realigned against a constructed reference of pseudo circRNA using HISAT2 (v 2.1.0) [55], and the read pairs mapped concordantly across a 10 bp region of the junction site were considered as circRNA transcripts. Differential expression was conducted using edgeR [56] fed with BSJ reads of circRNA. To identify the DE circRNA we used a *p*-value (*p*) of <0.05 and a fold change (FC) of >2.

### 4.7. cDNA Synthesis and Quantitative PCR

The total RNA was reverse transcribed with random hexamers. Then Real-time PCR was conducted using divergent primers. We used the ΔCt method to normalize the data using GAPDH and HMBS as internal references. Finally, the fold changes were log transformed for visualization. All reactions were performed in triplicate for each biological group. Primer sequences are provided in Appendix A.

### 4.8. mRNA and miRNA Expression Analysis

For mRNA expression analysis, the raw reads were demultiplexed, quality checked, and trimmed as explained above. The resulting FASTQ files were subjected to CIRIquant to quantify mRNA expression. In brief, the reads were aligned to the reference genome hg19 using HISAT2, and the mapped reads were fed into StringTie [57] to reassemble the transcriptome and estimate gene expression. Differential expression analysis was conducted by EdgeR, with FDR < 0.05 and FC > 1.5 set to identify DE mRNA.

Analysis of miRNA expression was performed as part of a separate study [16]. The sequencing reads were quality controlled by FastQC (v0.11.8) (http://www.bioinformatics.babraham.ac.uk/projects/fastqc) (accessed on 23 October 2020), and adaptors were trimmed using Cutadapt (v2.10) (DOI:10.14806/ej.17.1.200) (accessed on 20 October 2020) before aligning to the human reference genome build hg19 using Bowtie2 (v2.4.1) [58]. Then, using the miRBase mature miRNA reference annotation, the reads aligned to the mature miRNAs were counted using htseq-count (v0.7.2).

### 4.9. Co-Expression Analysis

We used DE circRNAs and DE mRNAs for this analysis. To identify co-expressed pairs, we used the Pearson correlation test (one-sided) and reported pairs with a significant positive correlation by setting an adjusted *p*-value < 0.05. The circRNA-mRNA pairs were then used to generate the circRNA–miRNA–mRNA network.

### 4.10. Construction of circRNA–miRNA–mRNA Network

We downloaded the TargetScan Release 7.2 prediction (https://www.targetscan.org/vert_72/) (accessed on 25 October 2020), including conserved mRNA targets of conserved miRNA families [59] and intersected the predictions with the DE mRNA list to find miRNA associating with the DE mRNA and construct miRNA-mRNA interactions. We then determined miRNA binding sites within DE circRNAs using sequences of circRNAs from circAtlas 2.0 [60] and the miRanda prediction algorithm [61], to determine circRNA–miRNA interactions. Finally, the two interaction networks were intersected using the common miRNA; those pairs with a shared miRNA, provided the miRNA was expressed in the samples, were combined to construct the circRNA–miRNA–mRNA regulatory network.

To generate the subnetworks, we first ranked the nodes of the interaction networks using the topological analysis methods provided by CytoHubba [62], and then selected the top nodes as hub nodes to create the subnetworks.

We also applied Fisher’s exact test to assess the enrichment of miRNA in the regulatory networks related to psychiatric disorders, using a recent systematic review of differentially expressed miRNA in psychiatric patients compared to healthy controls. Table 1 and Table 2 demonstrate the test details for PD-OS and DD-OS, respectively.

### 4.11. Functional Enrichment Analysis

The gene ontology (GO) and pathway enrichment analyses were performed using ToppFun from the ToppGene Suite (https://toppgene.cchmc.org) (accessed on 25 October 2020) [63]. We input either circRNA parental genes or target genes and set all human genes as the reference for the analysis. An FDR < 0.05 was set as the threshold for statistical significance in determining significant terms.

### 4.12. Visualization

Plots and graphs were constructed using R coding (v3.5). Heatmaps were constructed using the heatmap.2 function from the gplots package (http://CRAN.R-project.org/package=gplots) (accessed on 25 October 2020) (v3.0.1). The volcano plots were constructed using the EnhancedVolcano package (v3.12) [64]. The interaction networks were generated with the Cytoscape tool [65]. The remaining plots and graphs were constructed using the ggplot2 package (v3.2.1) [66].

## Figures and Tables

**Figure 1 ijms-25-12459-f001:**
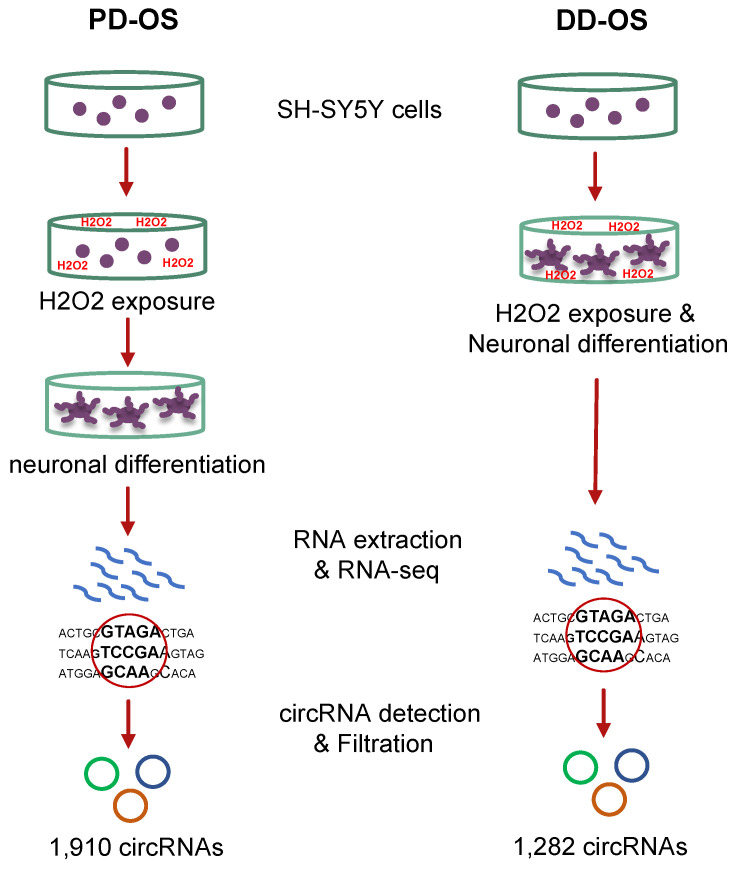
Schematic of the oxidative stress cell models in this study. The two oxidative stress models using SH-SY5Y neuroblastoma cells include: pre-differentiation oxidative stress (PD-OS), in which the oxidative condition was introduced before neuronal differentiation; and during-differentiation oxidative stress (DD-OS), in which the oxidative condition was introduced during differentiation. In the next step, the circRNA profile of each model was identified by RNA sequencing using total RNA depleted of ribosomal RNA, followed by bioinformatic analysis.

**Figure 2 ijms-25-12459-f002:**
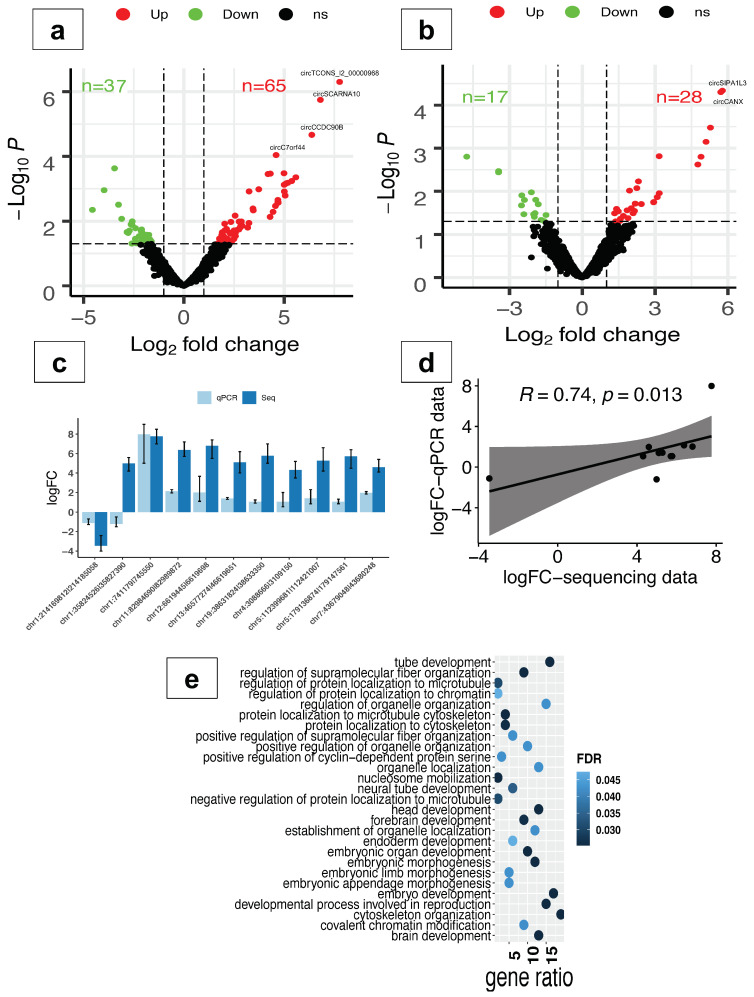
Analysis of circRNA differential expression in oxidative stress. Volcano plot generated using fold change values and *p*-values to illustrate the expression changes of circRNA in oxidative conditions PD-OS (**a**) and DD-OS (**b**). Red and green dots indicate significant upregulation and downregulation, respectively, with the vertical lines representing a fold change > 2 and the horizontal line representing a *p* < 0.05 cutoff. Differentially expressed circRNAs that survived multiple testing correction are labelled on plots. (**c**) Side-by-side comparison of the expression change from sequencing data and qRT-PCR results for the top 11 differentially expressed circRNAs. All reactions were performed in triplicate. (**d**) Correlation between the expression change from sequencing data and qRT-PCR results (Spearman’s rank, ρ = 0.74, *p* = 0.013). (**e**) Gene Ontology (GO) enrichment analysis for the altered circRNA parental genes in PD-OS. All the significantly enriched GO terms are shown (FDR < 0.05).

**Figure 3 ijms-25-12459-f003:**
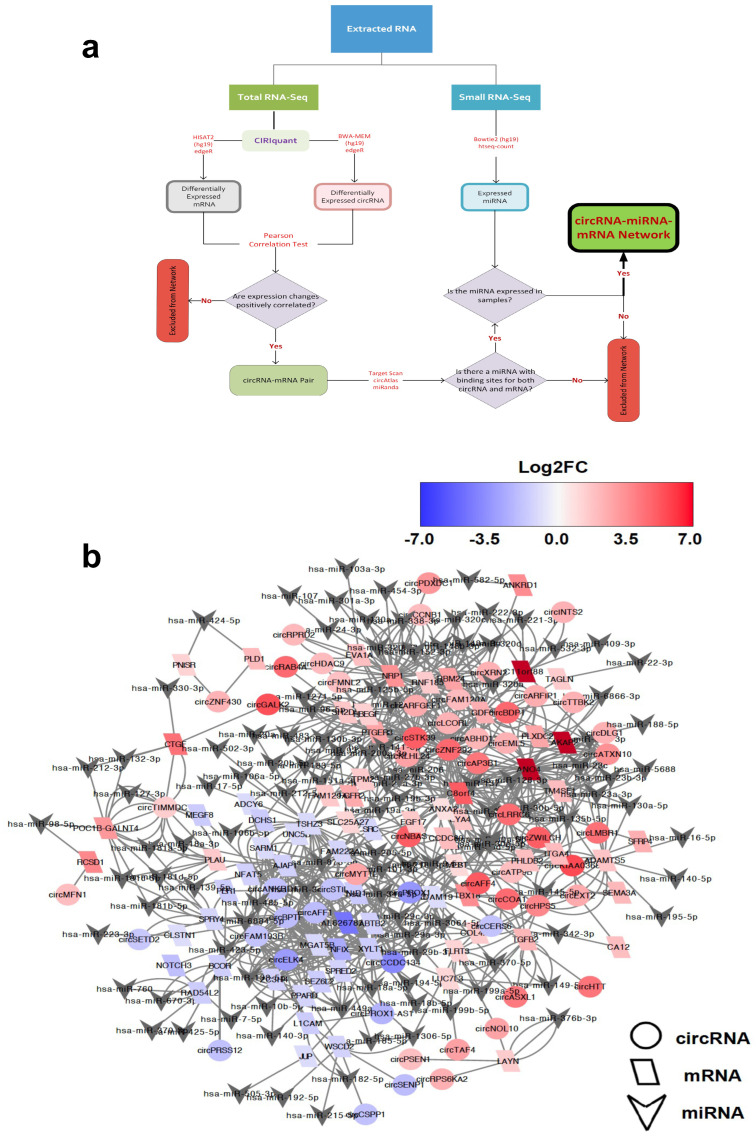
The circRNA-associated ceRNA network in oxidative stress. The potential interaction network of circRNA-miRNA-mRNA. (**a**) The networks are constructed based on the co-expression between the DE circRNAs and DE mRNAs and the predicted miRNA, with binding sites for each correlated circRNA–mRNA pair. (**b**) The predicted interaction network in PD-OS includes 454 axes and 244 nodes. (**c**) The predicted network in DD-OS includes 122 axes and 103 nodes. The circle and the parallelogram represent DE circRNA and DE mRNA, respectively, with red and blue denoting upregulation and downregulation, respectively. The colour intensity associates with the expression fold change. The gray inverted triangles indicate miRNA.

**Figure 4 ijms-25-12459-f004:**
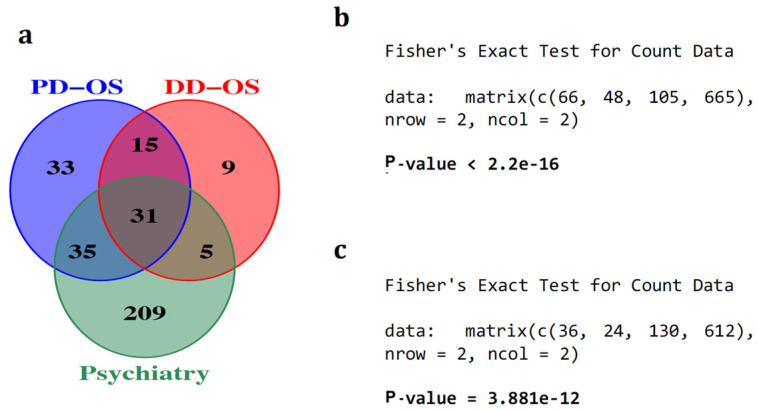
The enrichment of circRNA-sponged miRNAs in psychiatric disorders. (**a**) Of the 114 and 60 miRNAs that were predicted to be sponged by DE circRNA in PD-OS and DD-OS, respectively, more than 50% were dysregulated in psychiatric disorders. We observed significant enrichment of circRNA-sponged miRNAs in these diseases for both PD-OS (**b**) and DD-OS (**c**) conditions, using Fisher’s exact test.

**Figure 5 ijms-25-12459-f005:**
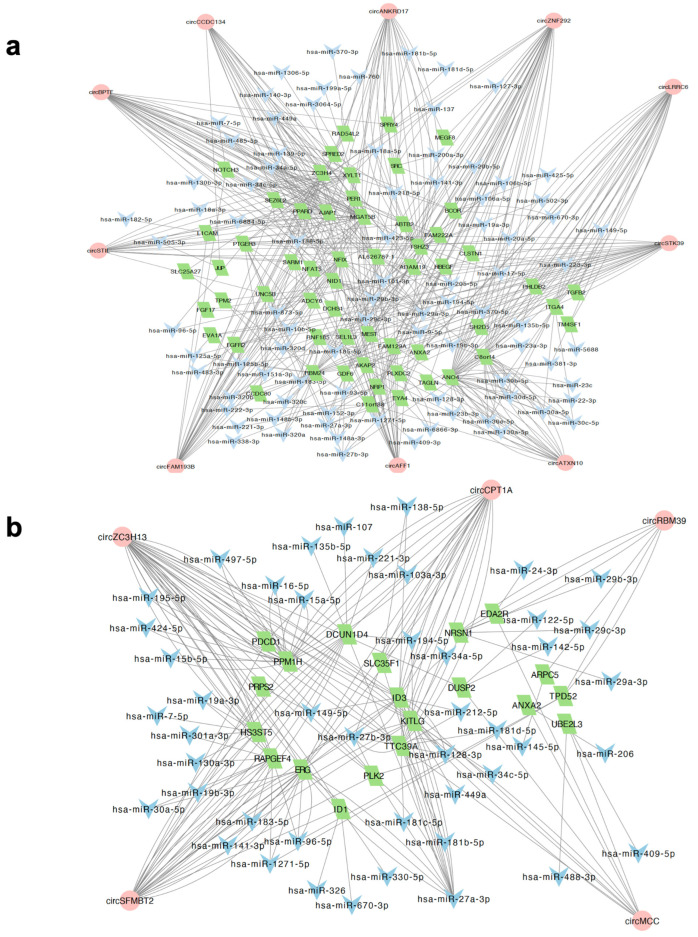
Identification of the ceRNA subnetwork. The interaction subnetwork in (**a**) PD-OS and (**b**) DD-OS. The subnetwork in PD-OS includes 10 hub circRNAs potentially regulating 57 mRNAs through interacting with 86 miRNAs. In the DD-OS subnetwork with 5 circRNA hubs, 20 mRNAs are regulated through 46 miRNAs. The circle represents circRNA, the parallelogram denotes mRNA, and the inverted triangle indicates miRNA.

**Figure 6 ijms-25-12459-f006:**
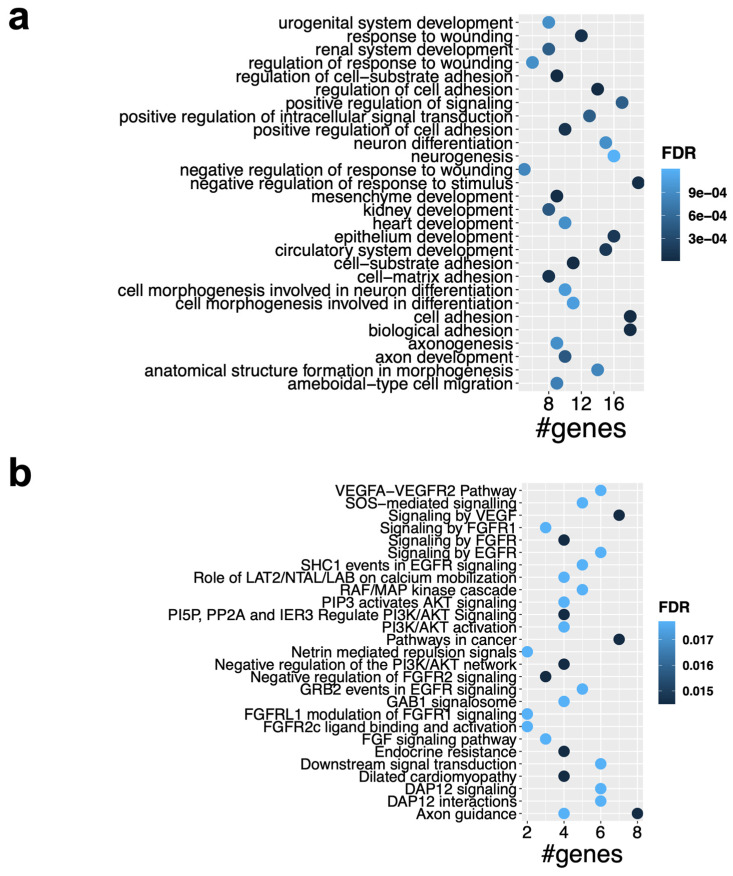
Functional analysis of the ceRNA subnetwork. (**a**) GO enrichment analysis for the target genes in the ceRNA subnetwork in PD-OS, which showed significantly enriched biological processes (FDR < 0.05). (**b**) Pathway enrichment analysis of the target genes in PD-OS, which indicated a significant enrichment of the genes in several pathways (FDR < 0.05). Selected GO terms and pathways are shown.

**Table 1 ijms-25-12459-t001:** Fisher’s exact test for examining the enrichment of miRNA within the PD-OS regulatory network in psychiatric diseases.

	miRNAs Sponged by circRNAs	miRNAs Not Sponged by circRNAs	Total
**miRNAs associated with psychiatry**	circRNA-sponged, Psy-associated miRNAs 66	Not sponged, Psy-associated miRNAs 105	Total number of Psy-associated miRNAs 171
**miRNAs unassociated with psychiatry**	circRNA-sponged, Psy-unassociated miRNAs 48	Not sponged, Psy-unassociated miRNAs 665	Total number of Psy-unassociated miRNAs 713
**Total**	Total number of circRNA-sponged miRNAs 114	Total number of miRNAs not sponged 770	**Total number of miRNAs expressed in the cell line** 884

fisher.test(matrix(c(66,48,105,665),nrow=2,ncol=2),alternative="greater")

**Table 2 ijms-25-12459-t002:** Fisher’s exact test for examining the enrichment of miRNA within the DD-OS regulatory network in psychiatric diseases.

	miRNAs Sponged by circRNAs	miRNAs Not Sponged by circRNAs	Total
**miRNAs associated with psychiatry**	circRNA-sponged, Psy-associated miRNAs 36	Not sponged, Psy-associated miRNAs 130	Total number of Psy-associated miRNAs 166
**miRNAs unassociated with psychiatry**	circRNA-sponged, Psy-unassociated miRNAs 24	Not sponged, Psy-unassociated miRNAs 612	Total number of Psy-unassociated miRNAs 636
**Total**	Total number of circRNA-sponged miRNAs 60	Total number of miRNAs not sponged 742	**Total number of miRNAs expressed in the cell line** 802

fisher.test(matrix(c(36,24,130,612),nrow=2,ncol=2),alternative="greater")

## Data Availability

Raw sequencing data for total RNA sequencing is available at the Gene Expression Omnibus (Accession Number: GSE161860). Raw sequencing data for small RNA sequencing is also available at the Gene Expression Omnibus (Accession Number: GSE182627).

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
