# Peer review of "Oxidative Stress-Associated Alteration of circRNA and Their ceRNA Network in Differentiating Neuroblasts"

_ijms, 2024, doi:10.3390/ijms252212459_

Round 1
Reviewer 1 Report (New Reviewer)
Comments and Suggestions for Authors
Minor Points:
- Abstract Section:
- The abstract is unclear and does not effectively communicate the key results or conclusions. It needs a clear progression from the background to the objectives, methods, results, and specific conclusions. This will provide clarity for readers on the study's outcomes.
- Formatting Issues:
- The document appears to have been uploaded in a track changes version, which should be resolved. Please ensure that track changes are accepted or rejected before final submission.
- There are references to specific lines such as line 579 and line 585. These references are not aligned with the manuscript and should either be corrected or removed.
- Graphs and Images:
- The images need better clarity, specifically by increasing the font size for readability. Additionally, ensure that all figures are properly formatted and described.
Major Points:
- Methods Section:
- The description of how oxidative stress was induced, and why certain models were chosen, should be expanded for clarity.
- What is ATRA, it has not been defined anywhere in the text
- Dose exposure to H2O2 is indicated but the results do not show this dose response. Please provide a rational for this dose treatment.
- Please provide a reference for RNA integrity number (RIN).
- Single Cell Line:
- The use of a single cell line (SH-SY5Y) limits the study's generalizability. It would be better to include additional cell lines to confirm the findings across different biological models. Using a cancer cell line doesn’t reflect the neural response to oxidative stress. Comparing the expression neural cells or brain slice cultures would provide a better understanding representation of the RNAs modulated during oxidative stress.
- The authors mention neural differentiation, SH-SY5Y neuroblastoma cells are cancer cells and true neural differentiation is not present in these cells. This makes the conclusions and discussions irrelevant to neural differentiation. Using differentiating neurons would give a true picture of the RNA network being investigated. The discussion will probably have to be refined to address cancer cells.
- A major drawback of this study is the use of a single cell line.
- ceRNA Network Analysis:
- The construction of the ceRNA network is well-presented but could benefit from functional validation. Experimental support for the proposed networks or should be discussed to strengthen the claims.
- Focus of the Study:
- The study seems somewhat scattered, particularly in the discussion. A stronger alignment between the research objectives, data, and conclusions would help improve the manuscript's cohesiveness.
Quality of English Language:
- Minor edits to the English language are needed to improve clarity and readability. The manuscript is well-written overall, but some sentences need rephrasing for clarity.
Comments on the Quality of English Language
Minor edits required
Author Response
Please see the attachment.

Reviewer 2 Report (New Reviewer)
Comments and Suggestions for Authors
In their paper entitled “Oxidative stress-associated alteration of circRNA and their ceRNA network in differentiating neuroblasts”, the Authors report the results of a study aimed at understanding the regulatory system that mediate and mitigate oxidative stress before or during neural differentiation.
The reported study provides insight into RNA-mediated regulatory processes that might link stress exposure to neurodevelopmental disorders.
The paper is of interest and suitable for International Journal of Molecular Sciences.
I only have one main suggestion for the Authors:
Interestingly, some papers reported that the sponge action of circRNA can be also played on RNA-binding proteins (RBPs). Given the central role of RBPs in the regulation of mRNA translation, especially in the Nervous System, where this level of regulation is even involved in learning and memory, I suggest to cite this phenomenon in the “Discussion” section.
Author Response
Please see the attachment

This manuscript is a resubmission of an earlier submission. The following is a list of the peer review reports and author responses from that submission.
Round 1
Reviewer 1 Report
Comments and Suggestions for Authors
The article entitled “Oxidative stress-associated alteration of circRNA and their ceRNA network in differentiating neuroblasts” aims to investigate the close correlation between oxidative stress and circRNA, through the use of a clear and very detailed strategy.
The authors provided comprehensive novel information about the involvement of circRNA-mediated network in oxidative stress pathways during neuronal differentiation, exploring how circRNAs are differentially expressed under oxidative stress conditions.
It is extremely interesting how the authors explore new insight into the molecular effects of the oxidative stress pathway, a complex cellular process involved in a wide range of physiopathological conditions, starting from aging to neurodegenerative disorders.
The authors also provided important insights into the possible study of these molecular mechanisms in various diseases and therapeutic strategies, e.g. regenerative medicine and cancer therapy.
However, the limitations of the present study should be more thoroughly evaluated.
The references reported are appropriate and include recent literature data, in line with the present study.
The resolution of Figure 3 and Figure 5 is too poor.
In Figure 5, choosing different colors from those used could improve the quality of the figure and make it easier to read.
Reviewer 2 Report
Comments and Suggestions for Authors
Mahmoudi et al investigates the alterations in the expression of circRNAs, miRNAs and mRNAs in differentiating neuroblasts prior to or during differentiation upon the induction of oxidative stress by hydrogen peroxide. To this end, the authors perform RNA sequencing with total RNAs extracted from cells treated with hydrogen peroxide during or prior to differentiation. Using in silico approaches, the authors then establish ceRNA networks from differentially expressed (DE) RNAs.
The authors provide a transcriptome-wide account of DE transcripts in differentiating neuroblasts upon the induction of oxidative stress. Although the resulting data could be of interest to the researchers working in the field, there are some limitations that should be addressed prior to considering publication.
Major points:
1. Figure 2: (a) Each component of the figure should be labeled with a different alphabet (thus, a b, c, d). (b) Resolution should be improved, especially for panels a and c. (c) What is the rationale behind performing a GO analysis with the cognate transcripts considering the diverse functions of circRNAs? Any evidence for candidate circRNAs regulating their cognate mRNAs? (d) I do not see much of a correlation between the RNA-seq and qRT-PCR data presented in Figure 2b. I believe that it would be better to present the data in a bar graph (RNA-seq and qPCR data side-by-side). (e ) the authors should validate DE circRNAs upon treatment of total RNAs with RNAs R to eliminate potential contribution from linear transcripts.
2. Line 122: It is highly surprising that the authors report only a single DE circRNA between the two conditions. Any biological explanation for this?
3. Lines 140-155: (a) Did the authors use only DE circRNAs, DE miRNAs and DE mRNAs to construct the ceRNA network? It is difficult to follow through the text. I advise preparing a chart similar to Figure 1 to summarize the filtering or analysis strategy. (b) More importantly, the authors should select a circRNA-miRNA-mRNA network and validate functionally by reverse genetics approaches. For example, it would be nice knockdown one of the circRNAs and check the expression of relevant miRNA and mRNAs, in addition to tracing potential phenotypic differences.
4. I am not quite convinced that the same sequencing data could be used for the reliable analysis of circRNAs, miRNAs, and mRNAs, each having a different size and structure. Typically, small RNA selection is employed for the enrichment of miRNAs while RNAse R treatment is almost a pre-requisite for the analysis of circRNAs. Considering the fact that the ceRNA networks are not functionally validated, I am afraid that the sequencing strategy used in this study raises concerns about the reliability of the in silico results.
5. Fig 3 and 5: Resolution should be improved.
6. Methods: (a) RNA-seq of circRNAs: Typically circRNA-seq library construction involves treatment of RNAs with RNAse R to eliminate linear RNAs to enrich for circRNAs. Why did the authors employ ribodepletion only? (b) Why use hg19, instead of hg38, for mapping the reads to the reference genome? (c ) the RT-qPCR primer sequences should be listed in a Table. Also, the qRT-PCR protocol should be included in Methods section. (d) DR miRNA and mRNAs should also be validated by qPCR.
Minor points:
1. Line 381: “too” should be “to”
Comments on the Quality of English LanguageMinor editing would help.
Round 2
Reviewer 2 Report
Comments and Suggestions for Authors
The authors address all the points raised by the reviewers. However, the explanations generate more concerns.
1. The RNA-seq data is already published by the same group (PMID: 33276438) and the authors avoid citing this work properly in Methods. Along the same line, apparently the small RNA-seq data is used for another publication, which is also not acknowledged. I was unable to verify the small RNA-seq data since the authors do not share the accession number.
2. Figure 2C: The number of replicates in qRT-PCR experiments is unknown. Also, SDs are missing in the bar graph.
3. Figure 3, 5 and 6 are still blurry.
4. Without any experimental validation of a potential ceRNA network important for oxidative stress response during neuronal differentiation, the current form of the manuscript does not provide sufficient progress to warrant publication, especially considering the fact that both RNA-seq data have been already used elsewhere.
Comments on the Quality of English LanguageMinor corrections are sufficient (superscripts..etc).
